# Linking genetic and phenotypic bedaquiline resistance in *Mycobacterium tuberculosis* strains from Georgia

Nino Maghradze[1,2,3], Chloé Loiseau[1,2], Galo Adrian Goig[1,2], Nino Bablishvili[3], Levan Jugheli[1,2], Sonia Borrell[1,2], Nestani Tukvadze[1,2,3], Russell Ryan Kempker[4,5], Zaza Avaliani[3], Sebastien Gagneux[1,2]*

**1** Swiss Tropical and Public Health Institute, Basel, Switzerland, **2** University of Basel, Basel, Switzerland, **3** National Center for Tuberculosis and Lung Diseases (NCTLD), Tbilisi, Georgia, **4** Division of Infectious Diseases, Department of Medicine, Emory University School of Medicine, Atlanta, Georgia, United States of America, **5** Departments of Epidemiology and Global Health, Rollins School of Public Health of Emory University, Atlanta, Georgia, United States of America

* sebastien.gagneux@swisstph.ch

## Abstract

### Introduction

Resistance to bedaquiline – a novel, promising medication against tuberculosis (TB), is already emerging, and uncertainties regarding the role of the different resistance-conferring mutations complicate the development of molecular diagnostic tools for detecting resistance. Mutations in the three genes *atpE, pepQ, and Rv0678* have been associated with increased minimum inhibitory concentrations (MICs) to bedaquiline in *Mycobacterium tuberculosis* (Mtb). Here, we studied the effect of known and novel mutations in these genes on the phenotypic susceptibility to bedaquiline in Mtb isolates from patients with drug-resistant TB in the country of Georgia.

### Methods

We used retrospective Mtb isolates (2011–2019) with whole-genome sequencing data, and prospectively collected diagnostic isolates with phenotypic resistance (2019–2022) to bedaquiline at the Georgian National Reference Laboratory. We determined bedaquiline MIC values using the Sensititre™ MYCOTB MIC plate. MIC of 0.12 µg/mL was defined as borderline and MIC ≥ 0.25 µg/mL as a resistant isolate. A phylogeny was inferred to assess the likely role of the identified variants in bedaquiline resistance, while taking into consideration population structure of the strains analyzed.

**Data availability statement:** All relevant data are within the paper and its Supporting Information files (S1), including only accession numbers for the genomes published at https://www.ebi.ac.uk/ena/browser/home, no additional potentially identifiable data is included in the S1 supplementary material.

**Funding:** This work was supported by the European Research Council (ECOEVODRTB Grant agreement ID: 883582) - Sebastien Gagneux and Swiss National Science Foundation (grant number 320030-227432). The funders had no role in study design, data collection and analysis, decision to publish, or preparation of the manuscript. https://erc.europa.eu/homepage

**Competing interests:** The authors have declared that no competing interests exist.

## Results

We analyzed a total of 69 Mtb isolates and identified 61 mutations across the three target genes. Seventeen (27.8%) of these variants were associated with borderline (0.12 µg/mL) or resistant (≥0.25 µg/mL) MICs to bedaquiline. In addition to six previously described bedaquiline resistance-conferring mutations in *atpE* and *Rv0678*, we identified two novel variants in *Rv0678* (Leu95Ser and Ile108fs) likely involved in bedaquiline resistance. We found a Tyr92Cys mutation in *Rv0678* in two epidemiologically linked isolates, which likely emerged as a consequence of previous exposure to clofazimine.

## Conclusion

Consistent with previous reports, our study confirms that mutations in *Rv0678* are the most frequent cause of bedaquiline resistance in Georgia, in addition to an increasing clinical relevance of mutations in *atpE,* while the role of *pepQ* mutations remains to be defined.

## Introduction

During the past decade, a number of novel anti-tuberculosis drugs have been approved for the treatment of multidrug-resistant (MDR) and extensively drug-resistant (XDR) tuberculosis. This includes bedaquiline (BDQ), a novel class of anti-mycobacterial medication with a high potency [1] that has led to substantially improved treatment outcomes among persons with drug-resistant tuberculosis (DR-TB) [2]. Since 2022, the World Health Organization (WHO) recommends BDQ-based, all-oral and shorter treatment regimens for MDR-TB [3]. However, acquired resistance to BDQ has already been reported and is emerging in several clinical settings [4–7]. Moreover, a study in treatment-naïve TB patients identified > 100 variants across three target genes in *Mycobacterium tuberculosis* (Mtb) that were linked to BDQ phenotypic resistance [5].

BDQ inhibits the mycobacterial ATP synthase [1], leading to ATP depletion, and cell death [8]. Although the direct resistance mechanism to BDQ selected *in vitro* usually involves the gene *atpE*, which encodes the drug target in the C subunit of the ATP synthase, efflux-pump overexpression is the most frequent cause of reduced susceptibility to BDQ in clinical Mtb isolates [4,9]. Specifically, inactivating mutations in the repressor gene *Rv0678* lead to an overexpression of *MmpS5-MmpL5*, which encodes an efflux pump, and can also lead to cross resistance to clofazimine [4,10]. In addition, *pepQ*, a gene encoding an aminopeptidase has been associated with increased MIC to BDQ and resistance to clofazimine [11]. Given that clofazimine has already been in use for MDR/XDR-TB for several decades, cross-resistance to BDQ resulting from previous clofazimine exposure also needs to be considered [12,13].

While data on BDQ resistance-conferring mutations is slowly accumulating [14–16], many uncertainties with respect to the specific role of the different mutations remain. Compared to the first edition of the WHO Mutation Catalogue where no BDQ resistance-associated mutation was included [17], the second edition published in 2023

[18] includes 70 BDQ resistance-associated variants, specifically defined as "Associated with Resistance" and "Associated with Resistance-interim". Out of these, six are in *atpE*, eight in *pepQ,* and 56 in *Rv0678* [18]. In addition, more than 200 variants in these three target genes were categorized as of "uncertain significance", which highlights the need for further research. Clarifying these, as well as potentially new mutations, is urgently required for improving molecular diagnostic tools [15].

Georgia programmatically implemented shorter treatment regimens, which include BDQ and delamanid as a backbone for DR-TB treatment in 2019. This intervention led to an increase in the success rate of MDR-TB treatment from 56% in 2016 to 72% in 2019 (National TB surveillance program, unpublished data) [19]. However, phenotypic resistance to these promising new drugs is already emerging [7], narrowing the options for the treatment of MDR-TB and XDR-TB and hindering further improvement in treatment success [20]. To enhance our understanding of the link between the Mtb genomic variation and BDQ resistance, we assessed the influence of genetic variants in the three target genes – *atpE, pepQ* and *Rv0678*, on phenotypic BDQ susceptibility in Mtb clinical strains from the country of Georgia.

## Materials and methods

### Ethics

The study was conducted as a part of the nationwide genomic epidemiological study of TB transmission at the National Reference Laboratory (NRL) of the National Center of Tuberculosis and Lung Diseases, in Tbilisi, Georgia, in collaboration with the Swiss Tropical and Public Health Institute (Swiss TPH) funded by European Research Council (ERC). Ethical approval was obtained by the relevant authorities in Georgia (the Institutional Review Board of the National Center for Tuberculosis and Lung Diseases, Tbilisi, Georgia) and Switzerland (Ethikkommission Nordwest- und Zentralschweiz). Obtaining individual written informed consent from each TB patient was waived by the Georgian Ministry of Health, both for the prospective as well as the retrospective TB patients included in this population-based study. All participant clinical and sample data were "pseudonymised" and the authors did not have access to any information that could identify individual participants at any stage during or after data collection.

### Mycobacterium tuberculosis isolates

**Mtb isolates selected retrospectively through whole genome data.** Through the collaborative project with Swiss TPH~8,000 diagnostic/baseline Mtb isolates have been sequenced and analyzed using an in-house pipeline [21,22]. From the resulting Mtb genomes, we identified all strains harboring any nonsynonymous mutations, frameshift, indels and nonsense mutations located in the following BDQ-resistance associated genes: *atpE, pepQ, and Rv0678,* dated from 01/01/2011–17/06/2019 (Swiss TPH MTBC genomic database). Only the genomic variants that met the following sequencing thresholds were considered: minimum mapping quality of 20, minimum read depth 7X, minimum base quality score of 20 and maximum strand bias 90%. The Mtb lineage of each strain was determined by interrogating genomic positions from a well-defined catalog of lineage-defining single nucleotide polymorphisms (SNPs) [23]. We included all the isolates with mutations irrespective of the allele frequency.

**Mtb isolates collected prospectively through phenotypic BDQ drug susceptibility testing.** Programmatic routine phenotypic drug susceptibility testing (DST) using BACTEC Mycobacterial Growth Indicator Tube (MGIT) on bedaquiline was implemented at the NCTLD NRL in June 2019. The WHO interim critical concentration for BDQ of 1 μg/mL [24] defining resistance was adopted into the routine second-line phenotypic DST. We identified diagnostic (baseline) BDQ-resistant isolates through the NRL laboratory database and retrieved the available samples from the NRL biobank from 17/06/2019–30/06/2020.

Clinical and bacteriological metadata – previous TB history, TB profile, treatment regimen and treatment outcomes, were accessed through the National Tuberculosis surveillance electronic database of the NCTLD on the stage of data analysis between 01/09/2023–29/02/2024.

## DNA extraction and sequencing

DNA was extracted for each isolate following the CTAB method [25] and Nanodrop measurements were used for evaluating the DNA concentration and purity. Sequencing was performed on an Illumina Novaseq 6000 instrument at the genomics core facility of the ETH Zurich and University of Basel in Switzerland.

## Bacteriology and MIC evaluation

Isolates were cultivated on Lowenstein-Jensen media until sufficient growth was observed. Sub-cultured isolates were used to make a suspension adjusted to McFarland standard 0.5, and 100ul of the suspensions were cultivated on Sensititre MycoTB MIC plates [26] with pre-prepared broth medium with a range of BDQ concentrations as follows: 0.03-0.06-0.12-0.25-0.5-1-2-4 µg/mL. The Sensititre MycoTB MIC (Catalog number: MYCOTB; Trek Diagnostic Systems, Cleveland, OH) is a commercially available dry microdilution plate, which was used for MIC testing at 99% inhibition, for all Mtb isolates included in our study. After inoculation, plates were covered with sealers and incubated at $37^0$C. One growth control was included per sample in 7H9 medium without BDQ. Each batch also included the fully susceptible strain H37Rv as a control. For validation, an Mtb isolate known to be resistant to BDQ, provided by the Supranational Reference Laboratory at the Institute of Tropical Medicine (ITM), Antwerp, Belgium for quality control panel, was also tested. We checked the plates for contamination after 24 and 48 hours. Cultures were monitored at 7, 10, 14 and 21 days by visual examination using a mirror viewer. For the purpose of result validation, MYCOTB testing was repeated for all Mtb isolates showing an increased MIC, in addition to repeated pDST for phenotypically resistant isolates.

## Phylogenetic inference

To construct a phylogenetic tree, we obtained an alignment of polymorphic positions by concatenating all high-quality SNPs in the dataset, excluding resistance-associated positions. Non-fixed SNPs (<90% allele frequency) or positions covered with less than 7 reads were encoded as "X" in the alignment. Positions in the alignment where more than 10% of sequences had an X, were discarded. The SNP alignment was used to infer a maximum-likelihood phylogeny using IQ-TREE 2 [27], with the general time-reversible model of sequence evolution, indicating the invariant sites of each nucleotide. The tree and corresponding metadata were visualized and plotted using ggtree [28] and ggtreeextra [29].

## Results

### Isolate selection

Group 1; based on putative genotypic resistance: from our Mtb genomic database covering the years 2009–2018, we retrospectively identified 71 Mtb isolates with at least one frameshift or non-synonymous mutations in *atpE, pepQ* and/or *Rv0678* genes (Fig 1). Seventeen (24%) of these samples were excluded due to a missing primary culture in the biobank, or a contamination during re-culturing (missing isolate – 12; contamination – 4; inefficient growth −1), leaving 54 (76%) isolates for further investigation.

The selection of strains included Mtb strains harboring genomic variants in atpE, pepQ or/and *Rv0678* genes from 2008 and 2019 and all available Mtb isolates previously tested as phenotypically resistant to BDQ. The total study isolates (N = 69) comprised the genomic variant dataset (2009–2018) and the phenotypically resistant isolates with available whole genome sequencing data.

Group 2; based on phenotypic resistance: from the NCTLD NRL database, we identified 36 Mtb isolates from June 2019 through January 2022 with BDQ resistance determined by standard phenotypic DST. WGS data was available for 15 (41.7%) of these Mtb isolates, which were therefore also included for further analysis (Fig 1).

Our final combined Mtb isolate set (Group 1 + Group 2) comprised a total of 69 isolates, including 54 isolates with known genomic variants potentially involved in BDQ resistance and 15 phenotypically BDQ-resistant isolates based on routine DST data, for which we also had genomic data available.

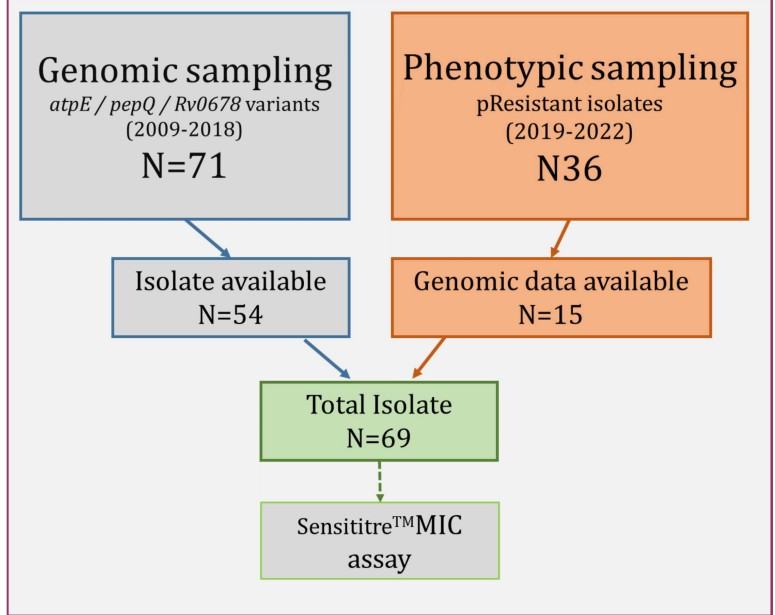

**Fig 1. Schematic representation of Mtb isolate selection.**

## Distribution of BDQ MICs across the study isolates

For the 69 Mtb isolates included in our study, we determined the BDQ MIC using the Sensititre MycoTB. The median BDQ MIC across all 69 isolates was 0.06 µg/mL[Inter quartile range: 0.03–0.012], with a maximum MIC value of 0.5 µg/mL. We defined Mtb isolates as BDQ susceptible at an MIC of ≤0.06 µg/mL, as borderline resistant at an MIC of =0.12 µg/mL, and as resistant at an MIC of >0.12 µg/mL.

**Susceptible isolates (MIC≤0.06 µg/mL) N = 46.** A total of 46/69 (66.7%) isolates had a BDQ MIC ≤ 0.06 µg/mL and were thus considered BDQ susceptible [25]. This included 3 of the 15 isolates originally classified as BDQ-resistant based on routine pDST using the MGIT, Bactec 960 system. Repeating the pDST confirmed the BDQ susceptibility of these isolates in addition to an absence of mutation in the target genes. Seven (15%) of the 46 BDQ-susceptible isolates had the Phe76Leu mutation in *atpE*. Another 24/46 isolates (52.2%) had one of 15 identified fixed mutation in *pepQ.* Of note, across our study, *pepQ* mutations were only observed in BDQ-susceptible isolates. Another 12/46 (26%) isolates had mutations in *Rv0678* despite of exhibiting MIC ≤ 0.06 µg/mL. Four of these had amino acid substitutions with uncertain significance described in the WHO Mutation Catalogue (Phe93Leu, Arg96Trp, Arg89Leu and Ser52Phe) [18], and the others were not previously described.

**Borderline resistant isolates (MIC = 0.12 µg/mL) N = 11.** A total of 11/69 (15.9%) isolates showed borderline resistance with a BDQ MIC of 0.12 mLµg/mL MIC. Eight of these 11 isolates (72.7%) had a mutation in *Rv0678*. This included Asp47fs, which has been associated with resistance and Phe93Leu and Asn70Ile, both with uncertain significance based on the second edition of the WHO Mutation Catalogue [18]. One substitution in *Rv0678* is newly reported here – Ala86Ser (Table 1). Additionally, one isolate showing borderline resistance, had a Glu163Asp substitution in *Rv0678* at a 4% allele frequency. This latter isolate, along with two isolates with no mutation in any of the three genes, were initially declared as BDQ susceptible by routine pDST. No fixed *atpE* or *pepQ* mutations were observed in this group of isolates (Table 1) (S1 Table).

**Table 1. Genomic variants associated with an increase of BDQ MIC ≥ 0.12 µg/mL.**

| atpE | pepQ | Rv0678 | #Isolates | MIC (µg/mL) |
|---|---|---|---|---|
| WT | WT | Ala86Ser | 1 | 0.12 |
| WT | WT | Asn70Ile | 1 | 0.12 |
| WT | WT | Asp47fs | 1 | 0.12 |
| WT | WT | Phe93Leu | 5 | 0.12 |
| WT | Glu360* [1] # | Leu95Ser, Ala102Asp# | 1 | 0.25 |
| WT | WT | Ile67fs # | 1 | 0.25 |
| WT | WT | Tyr92Cys | 2 | 0.25/0.5 |
| WT | WT | Ala102Asp | 1 | 0.5 |
| Ile66Met | WT | WT | 2 | 0.5 |
| WT | WT | Leu32Ser | 1 | 0.5 |
| Ala63Pro # | WT | Ile108fs # | 1 | 0.5 |

Legend: Data represents **mutations** within 17 isolates (excluding WT) exhibiting resistance using ECOFF cut-off, including borderline resistance. Additional mutations in genes with- # were detected in <90% of reads. fs –stands for the frameshift mutation; [1]premature stop codon; No pepQ mutant was associated with an increased MIC to BDQ. Mutations colored in red are included in the second edition of the WHO Mutation Catalogue as "associated with resistance", while mutations in blue are included as "uncertain significance" [18]. Mutations in black are newly reported here. For detailed information, refer to S2 Table.

**Resistant isolates (MIC = 0.25-0.5 µg/mLmL) N=12.** Twelve (17.4%) of the 69 isolates analyzed exhibited a BDQ MIC of 0.25 µg/mL or higher, reflecting BDQ resistance. Of these, 6 had an MIC of 0.25 µg/mL and 6 an MIC of 0.5 µg/mL. Three out of these 12 isolates (25%) had no mutation in any of the three target genes associated with BDQ resistance.

Among the nine isolates with mutations in the target genes, six had a mutation in *Rv0678*. This included five known variants - Tyr92Cys (Uncertain Significance), Leu95Ser (Uncertain Significance), Ala102Asp (Uncertain Significance) Leu32Ser (Associated with Resistance) and a frameshift mutation – Ile67fs (Associated with Resistance) that were included in the second edition of the WHO Mutation Catalogue [18]. Noteworthy, one isolate harbored two mutations, Ala63Pro (Associated with Resistance) in *atpE* and Ile108fs in *Rv0678*, both at 25% of allele frequency, suggesting that they were phased and therefore present in the genomes of the same bacterial subpopulation. Four variants with low allele frequencies – Arg38Gly, Ala59Glu, Ile108fs (Table 1) and Met139Thr (S2 Table) detected in our dataset – are reported in BDQ-resistant strains for the first time here.

Additionally, two out of the nine isolates harbored the previously described Ile66Met mutation in *atpE,* which is also included in the second version of the WHO Mutation Catalogue. No *pepQ* mutation was observed in this group of BDQ-resistant isolates. Detailed information for all mutations and corresponding MIC data is included in S2 Table.

### Clinical metadata for BDQ resistant (MIC ≥ 0.25 µg/mL) Mtb isolates

From the 12 Mtb isolates with BDQ resistance (i.e., exhibiting a BDQ MIC value ≥0.25 µg/mL), full or partial second line pDST data was available for all patients. Of these, two (16.6%) classified as poly-resistant (resistant to isoniazid and streptomycin), five (41.7%) as MDR, and five (41.7%) as pre-XDR cases (defined by WHO as MDR with additional resistance to fluoroquinolones) (Table 2). Five (41.7%) of the 12 patients had experienced a previous TB episode, but the previous treatment regimens used during these episodes are unknown.

Out of these 12 patients with BDQ resistance, only 3 (25%) were declared as cured, while six (50%) had unfavorable treatment outcomes – i.e., failure or death. The Mtb isolates from the latter group of patients harbored mutations in either

**Table 2. Phenotypic, genomic and epidemiological data for the patients with BDQ-resistant Mtb isolates (*MIC ≥ 0.25 µg/mL*).**

| Phenotypic data | | | | | | | | | Genotypic data | | | Epidata | |
|---|---|---|---|---|---|---|---|---|---|---|---|---|---|
| ID | TB profile # | Phenotypic Drug Susceptibility | | | | | | | *atpE* | *Rv0678** | TB history** | Treatment regimen*** | Outcome |
| | | Mfx[1] | Mfx[2] | Lfx | BDQ | Lzd | Cfz | Amk | | | | | |
| 1 | XDR | R | S | - | R | - | - | - | WT | WT | Yes | Cs Cfz Lzd BDQ Dld | - |
| 2 | MDR | S | S | S | R | - | S | S | WT | WT | No | - | Cured |
| 3 | MDR | S | S | S | R | - | S | R | WT | WT | - | - | - |
| 4 | MDR | S | S | S | R | - | S | S | WT | WT | No | Lfx, Cs, Cfz, Lzd, BDQ | Cured |
| 5 | MDR | R | R | R | - | - | - | - | WT | Leu95Ser, Ala102Asp; | Yes | Mfx, PAS, Cm, Pto, Cs, Cfz | Death |
| 6 | SHE | S | S | S | - | - | - | - | WT | Tyr92Cys | No | Lfx, Eto, Pto, Pz, PAS, Cs, | Failure |
| 7 | XDR | R | R | R | R | - | S | S | Ile66Met | WT | No | - | Death |
| 8 | XDR | R | S | R | R | S | R | S | WT | Ala102Asp | No | Lfx Cfz Lzd BDQ | Failure |
| 9 | XDR | R | R | R | R | - | S | S | Ile66Met | WT | Yes | Lfx Cs Cfz Lzd BDQ | Failure |
| 10 | XDR | R | S | - | R | - | - | - | WT | Leu32Ser | Yes | Cs Cfz Lzd BDQ Dld | - |
| 11 | MDR | S | S | S | R | - | S | S | Ala63Pro[2] | Ile108fs[2] | No | Lfx Cs Cfz Lzd BDQ Dld | Cured |
| 12 | SHE | - | - | - | - | - | - | - | WT | Tyr92Cys; Glu451fs | Yes | - | Death |

\# Classification is based on updated WHO definitions.

\* Phenotypic DST was performed on MGIT (Bactec 960): [1] Mfx – moxifloxacin concentration 0.25 µg/mL; [2]Mfx concentration 1.0 µg/mL; Lfx – levofloxacin, BDQ – bedaquiline, Lzd – linezolid, Cfz – clofazimine and Amk – amikacin concentrations on MGIT – 1.0 µg/mL.

\*\* TB history denotes presence of active disease episode in past years.

\*\*\* Treatment regimen indicated in the table is primary treatment medications started before pDST result availability.

"-" denotes the missing information.

*atpE* or *Rv0678* (Table 2). Noteworthy, both isolates with the Ile66Met substitution in *atpE* and two isolates with the Tyr-92Cys change in *Rv0678* were linked to an unsuccessful outcome.

## Phylogenetic inference

A phylogenetic analysis of the 69 Mtb isolates included in our study revealed the following Mtb lineage distribution: 39 (56%) isolates belonged to lineage 2, 27 (40%) to lineage 4, and one (1%) to lineage 3 (Fig 2). Two isolates (3%) were identified as a mixture of lineage 2 and 4, latter being excluded from the phylogenetic tree.

Figure represents the phylogeny of 67 isolates (excluding two mixed lineage isolates (L2/L4)) from the study cohort, indicating isolate lineage, DR profile defined by genotypic drug susceptibility based on WHO definition (2021). Variants are stratified by genes – *atpE, pepQ* and *Rv0678*, corresponding MIC values are included in gradient.

Although a few of the *atpE, pepQ* and *Rv0678* variants were detected in two or more isolates, none were shared across different clades. However, given the limited sample size, and the fact that the two mutations identified in our dataset (Tyr-92Cys and Ile66Met) were found in epidemiologically linked cases (Fig 2), we cannot rule out the possibility of homoplasy or conclude that these mutations do not influence BDQ resistance.

## Discussion

This study explored the genotypic variants in the three known target genes of BDQ resistance - *atpE, pepQ, Rv0678*, and their influence on BDQ MICs in Mtb clinical isolates from the country of Georgia. Our results show that in Georgia, most BDQ resistance-conferring mutations occur in *Rv0678*. This included previously described as well as two novel resistance-associated mutations. Previously described mutations were also observed in *atpE*, including Ala63Pro and

Ile66Met, with the latter occurring in the context of two instances of treatment failure. In our dataset, no case of BDQ resistance were detected with *pepQ* mutations.

Our finding that most BDQ resistance-conferring mutations occurred in *Rv0678* is consistent with previous reports [4,7,9,14]. Six of the mutations we observed in *Rv0678* were previously described and recently included in the second edition of the WHO Mutation Catalogue [18]. Moreover, we identified two new mutations in *Rv0678* that were associated with an increased MIC to BDQ: Ile108fs (in combination with known Ala63Pro in *atpE*) and Ala86Ser. Noteworthy, there were two patients infected with Mtb isolates carrying the known *Rv0678* Tyr92Cys mutation, who failed their treatment. With clear epidemiological link (Fig 2), these two patients were sampled before 2015, having a history of previous TB, during which they are likely to have received clofazimine. Although the association between clofazimine therapy and bedaquiline resistance has been described in several studies [10,16], from our dataset, we could not confirm cross-resistance phenomena, as for the above mentioned patients the history of clofazimine exposure is unknown. Altogether, four out of six patients infected with Mtb isolates carrying *Rv0678* mutations putatively associated with BDQ resistance failed their treatment. This emphasizes the role of BDQ resistance in poor treatment outcomes [20].

The comparably low clinical frequency of *atpE* mutations involved in BDQ resistance contrasts with the high frequency of these mutations among *in vitro* selected mutants [13,14,30]. It has been suggested that *atpE* mutations suffer from a high fitness cost and are therefore selected against *in vivo* [30]. However, these mutations might serve as a key contributor to high-level resistance against bedaquiline.

In support of this view, several clinically relevant BDQ resistance-conferring mutations in *atpE* have been described and have also been added to the latest edition of the WHO Mutation Catalogue [18]. This includes *atpE* Ala63Pro and Ile-66Met, which we found in Mtb isolates from two patients, both of whom failed their treatment. Whether the *atpE* mutation

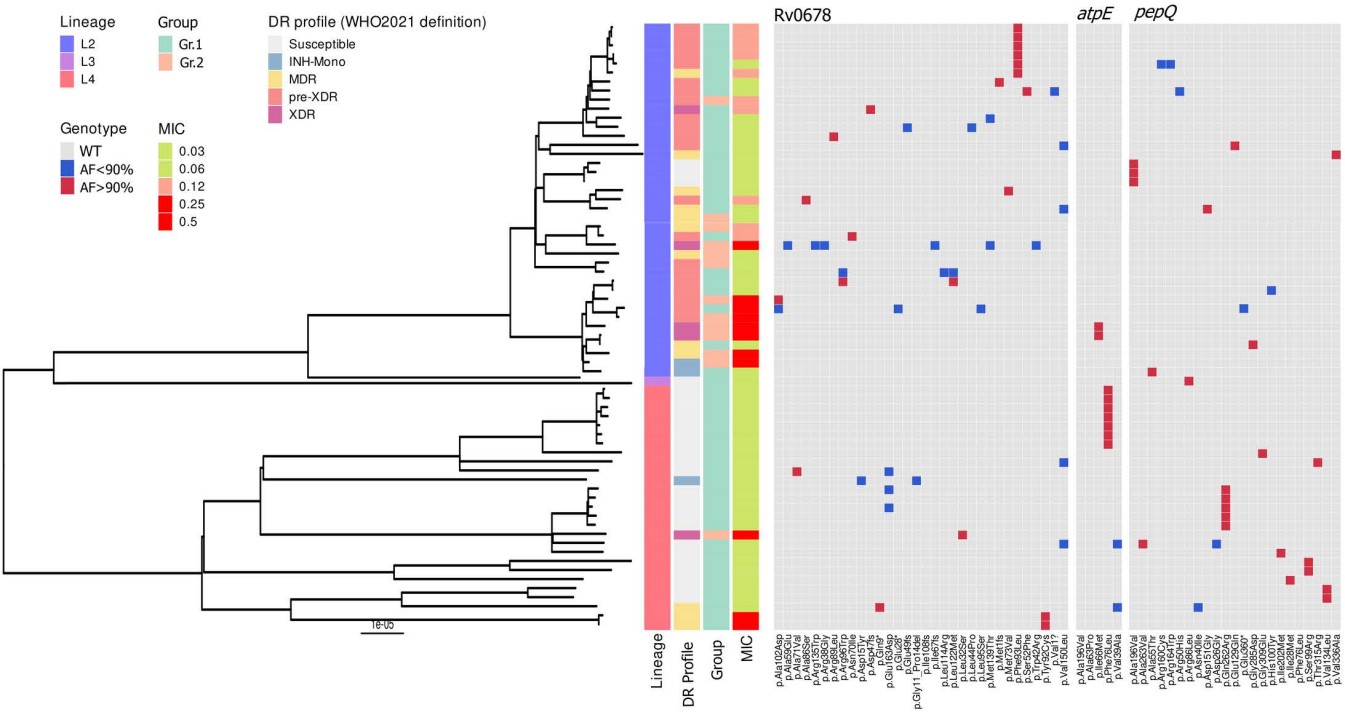

**Fig 2. Phylogeny of the study isolates with identified mutations, TB profile, lineage and corresponding MIC values.**

contributed to this treatment failure is difficult to ascertain, as both patients also showed evidence for acquired resistance to fluoroquinolone, which is known to be associated with treatment failure [20].

Among the 15 patients previously identified as BDQ resistant through routine phenotypic testing, six turned out to be BDQ susceptible based on repeated testing using both the Sensititre MycoTB assay and standard pDST, as well as based on the genotypic data generated during our study. These discrepant results highlight the current challenges of performing routine drug susceptibility testing for BDQ under programmatic conditions based on the current standards [31], with important consequences for adequate patient management. Methods to detect BDQ resistance (and susceptibility) more reliably will likely improve with more data accumulating over time.

Our study is limited due to the small sample size, which precluded formal statistical tests to explore more thoroughly the association between the different Mtb genetic variants and BDQ resistance versus susceptibility. This was particularly true for the new variants described in *Rv0678*.

In conclusion, the work presented here confirms the relevance of *Rv0678* mutations in the growing burden of BDQ resistance in Georgia and globally [7]. While we also observed *atpE* mutations leading to BDQ resistance in our dataset, we did not find BDQ-resistant cases due to *pepQ* mutations. Combining the data generated here with larger datasets in the future will increase our capacity to detect BDQ susceptibility and resistance more reliably, and thereby help preserve this valuable new drug that took many years to develop.

## Supporting information

**S1. Table. List of mutations in atpE, pepQ and *Rv0678*, isolate quantity, corresponding MICs and lineage distribution of study isolates.**
(DOCX)

**S2 Table. Study isolates (69) with corresponding accession number of the European Nucleotide Archive (ENA), mutations with variant frequency, inclusion in WHO catalogue (second edition) and MIC values.**
(DOCX)

**S3 Questionnaire. Inclusivity in global research questionnaire.**
(DOCX)

## Author contributions

**Conceptualization:** Nino Maghradze, Russell Ryan Kempker, Sebastien Gagneux.

**Data curation:** Nino Maghradze, Chloé Loiseau, Galo Adrian Goig, Levan Jugheli, Sonia Borrell, Sebastien Gagneux.

**Formal analysis:** Nino Maghradze, Chloé Loiseau, Galo Adrian Goig, Levan Jugheli.

**Funding acquisition:** Nestani Tukvadze, Russell Ryan Kempker, Zaza Avaliani, Sebastien Gagneux.

**Investigation:** Nino Maghradze.

**Methodology:** Nino Maghradze, Chloé Loiseau, Galo Adrian Goig, Nino Bablishvili, Levan Jugheli.

**Project administration:** Nino Maghradze, Nestani Tukvadze, Zaza Avaliani.

**Supervision:** Levan Jugheli, Sonia Borrell, Nestani Tukvadze, Russell Ryan Kempker, Sebastien Gagneux.

**Validation:** Chloé Loiseau, Galo Adrian Goig, Nino Bablishvili, Levan Jugheli, Sonia Borrell, Sebastien Gagneux.

**Writing – original draft:** Nino Maghradze.

**Writing – review & editing:** Nino Maghradze, Chloé Loiseau, Galo Adrian Goig, Levan Jugheli, Sonia Borrell, Nestani Tukvadze, Russell Ryan Kempker, Sebastien Gagneux.

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
