## [Decision Letter · Decision Letter 0]

PONE-D-24-38581Linking genetic and phenotypic bedaquiline resistance in Mycobacterium tuberculosis strains from GeorgiaPLOS ONE

Dear Dr. Gagneux,

Thank you for submitting your manuscript to PLOS ONE. After careful consideration, we feel that it has merit but does not fully meet PLOS ONE’s publication criteria as it currently stands. Therefore, we invite you to submit a revised version of the manuscript that addresses the points raised during the review process. Please submit your revised manuscript by Apr 14 2025 11:59PM. If you will need more time than this to complete your revisions, please reply to this message or contact the journal office at plosone@plos.org . Please include the following items when submitting your revised manuscript:

We look forward to receiving your revised manuscript.

Kind regards,

Atul Vashist, PhD

Academic Editor

PLOS ONE

4. We note that there is identifying data in the Supporting Information file < S1 Appendix.docx>. Due to the inclusion of these potentially identifying data, we have removed this file from your file inventory. Prior to sharing human research participant data, authors should consult with an ethics committee to ensure data are shared in accordance with participant consent and all applicable local laws.

-Location data

Please remove or anonymize all personal information, ensure that the data shared are in accordance with participant consent, and re-upload a fully anonymized data set. Please note that spreadsheet columns with personal information must be removed and not hidden as all hidden columns will appear in the published file.

Reviewers' comments:

Reviewer's Responses to Questions

**Comments to the Author**

1. Is the manuscript technically sound, and do the data support the conclusions?

Reviewer #1: Yes

Reviewer #2: Yes

Reviewer #3: Yes

2. Has the statistical analysis been performed appropriately and rigorously? 

Reviewer #1: N/A

Reviewer #2: Yes

Reviewer #3: N/A

3. Have the authors made all data underlying the findings in their manuscript fully available?

Reviewer #1: Yes

Reviewer #2: Yes

Reviewer #3: Yes

4. Is the manuscript presented in an intelligible fashion and written in standard English?

Reviewer #1: Yes

Reviewer #2: Yes

Reviewer #3: Yes

5. Review Comments to the Author

Reviewer #1: The above manuscript is well written and adds to the currently emerging BDQ resistance literature. The study shows that among the isolates from Georgia, BDQ resistance was mainly due to the mutations in Rv0678 supporting the other reports elsewhere and WHO. It also introduces the concern that the same mutations in some isolates might lead to phenotypic resistance and some may not.

Minor comments:

please make sure everywhere in the manuscript units of measurement are properly mentioned.

for example, micron symbol should be of universal symbol, microgram is not 'yg' (page 10),

missing units for MIC.

Page 7. please mention the reference to the literature for your MIC baseline. was the MIC defined at 90% or 99%? line 197: Sensititre mycoTB: please mention the catalogue number, company, country.

line 199: when classifying TB resistance wondering what you classify samples with MIC >0.06 but <0.12?

Line 204: please define routine? if done on sensititre or LJ medium or MGIT? how were the MIC cut off established? any references would be helpful since from the literature, it doesnt seem to clear on setting these cut offs. my personal concern also is with the pDST and the methods used differs so much that may be there is a need of a an strict universal pDST standard for BDQ.

line 216: REF? is reference missing?

line 224: Incomplete phrase missing information here: "Additional mutations in genes with- # were detected in <90% of reads".

lines 254, 255, 268: ul or ug?

Table 2: please mention BDQ MIC (ug/ml) in the MIC column.

Table 2: please provide reference for e MIC cut-off values for all different drugs mentioned here.

Line 320: do you mean the role of "Rv0678" and not "BDQ' mutations in poor treatment outcome?

Fig. 1: please change to N=, as per universal standards. mentioning N36, N15 etc., gives the impression of sample number rather than total numbers.

Also please reconsider this figure design. it is understandable but the standard flow chart probably will be better.

Fig. 2 resolution is poor. Cant read anything on the X-axis.

Reviewer #2: The manuscript by Maghradze et al is a useful contribution to the field. It is well written, easy to follow and draws sound conclusions. Methods seem appropriate for the study. I only have some brief comments.

I wouldn’t say the main resistance mechanism in vitro is atpE. Whilst these mutants can be isolated, several studies report and over representation of rv0678 mutants as being the dominant mechanism in vitro. Maybe reporting that atpE is high level resistance is relevant?

In figure 2, maybe an additional indicating whether the strain is susceptible, intermediate or resistant to bedaquiline. There is the MIC data, and the are described in the text, but may make it easier to read

There are 9 strains in table 1 with MIC of 0.25 or 0.5, yet 12 mentioned in the text variants are identified. Of the three with no SNPs in target genes was WGS performed?

The link to patient data is really nice, showing the influence of these mutations on outcomes

I think the statement “If confirmed, this would be similar to the situation with katG, for which the mutation Ser315Thr is rarely observed in the laboratory due to its high fitness cost in vitro despite being, by far, the most clinically relevant isoniazid resistance-conferring mutation due to its high fitness in vivo” is misleading, and should be altered or removed. I would argue that the disconnect with S315T is because in the lab any LOF mutation will provide resistance katG yet these mutants are non-viable in host due to the loss of catalase function. S315T is predominant in vivo, because it is one of the few mutations that preserves catalase function whilst also preventing the activation of INH. Again, also see comment above about frequency of atpE mutations.

Reviewer #3: Bedaquiline (BDQ) is a novel antituberculosis agent, that is pivotal to the potent multidrug regimens for the treatment of drug-resistant tuberculosis (TB). Although the genes associated with BDQ resistance has been described, the mutations that are associated with increased minimum inhibitory concentrations (MICs) have not been fully characterized. As such, the manuscript by Maghradze et al., that describes genomic correlates of BDQ resistance, provides useful information of interest to the field. Overall, the methods and results are presented clearly. A few detailed comments are bulleted below:

• Small inconsistencies in reporting number of programmatically determined BDQ-R isolates that were reclassified as susceptible or borderline:

o In the Results section, reports 3/15 initially classified as resistant that had a MIC<0.06 and no mutations, resulting in them being reclassified as susceptible. Also reports 2/15 initially classified as phenotypically resistant with MIC=0.12 and no resistance mutations. So should be 5/15 (?)

o In Discussion reports 6/15 identified through phenotypic testing turned out to be BDQ susceptible “based on repeated testing using both... pDST, as well as based on the genotypic data”

Unclear where this 6th isolate comes from. Presumably it is the one with a mutation at 4% allele frequency, but they specifically did not include AF in their inclusion criteria for mutations, so this is incredibly unclear.

• 3 isolates found to have MIC=0.25-0.5 (classified as phenotypically resistant) but no mutation in target genes

o They only look in the target genes atpE, pepQ, Rv0678. While these are the genes previously associated with resistance, I’m curious as to why they didn’t choose to investigate mutations in other genes associated with the MOA of bedaquiline or with the efflux pump

• I understand why they would choose to only show treatment outcomes for those classified as phenotypically BDQR, however I think it could have strengthened the paper to demonstrate the difference in treatment outcomes between the resistance groups. In addition, as the authors note, the study and sample size preclude any meaningful discussion of clinical outcomes.

• States that six of the phenotypically BDQR patients had a previous TB episode, but reports the percentage as 46.2% (of 12 total patients, 6 would be 50%). Then in the table, only lists 5 patients with known TB history and 1 with missing data.

• Unclear if the 3 patients without known outcomes are still undergoing treatment, or if the data was truly unavailable, making it difficult to understand the cure rate.

• Included mixed-lineage isolates in their analyses and tree. This might not be as big of an issue for the BDQR mutation calling since they include multiple mutations at different allele frequencies for a few of their samples, but they should not be in the tree.

6. PLOS authors have the option to publish the peer review history of their article (what does this mean? ). If published, this will include your full peer review and any attached files.

**Do you want your identity to be public for this peer review?** For information about this choice, including consent withdrawal, please see our Privacy Policy .

Reviewer #1: **Yes: ** Priya Banada

Reviewer #2: **Yes: ** Matthew McNeil

Reviewer #3: No

---

## [Author Response · Author response to Decision Letter 1]

30 Apr 2025

Reviewer #1

The above manuscript is well written and adds to the currently emerging BDQ resistance literature. The study shows that among the isolates from Georgia, BDQ resistance was mainly due to the mutations in Rv0678 supporting the other reports elsewhere and WHO. It also introduces the concern that the same mutations in some isolates might lead to phenotypic resistance and some may not.

Minor comments:

please make sure everywhere in the manuscript units of measurement are properly mentioned. For example, micron symbol should be of universal symbol, microgram is not 'yg' (page 10), missing units for MIC.

Response: corrected ɥg to μg.

Page 7. please mention the reference to the literature for your MIC baseline. was the MIC defined at 90% or 99%?

Response: Reference is now mentioned at line 147 (26 in the reference list). The percentage was added.

line 197: Sensititre mycoTB: please mention the catalogue number, company, country.

Response: The details are noted in the methods section line 149. We added the catalog number which is noted as: MYCOTB.

line 199: when classifying TB resistance wondering what you classify samples with MIC >0.06 but <0.12?

Response: The tested concentrations for bedaquiline in our study were 0.03-0.06-0.12-0.25-0.5-1-2-4 μg/mL (i.e. 2-fold dilutions). Hence there was no concentration formally tested between 0.06 and 0.12. In case of growth detection at 0.12 μg/mL, the isolate was classified as borderline resistant. In case of growth only at 0.06 or lower concentrations, the isolate was considered susceptible.

Line 204: please define routine? if done on sensititre or LJ medium or MGIT? how were the MIC cut off established? any references would be helpful since from the literature, it doesnt seem to clear on setting these cut offs. my personal concern also is with the pDST and the methods used differs so much that may be there is a need of a an strict universal pDST standard for BDQ.

Response: The routine pDST is performed on MGIT, Bactec 960 systems at the National Reference Laboratory. The details, including relevant references are noted in the methods section line 125. As for the consensus pDST critical concentration for bedaquiline, the collated inconsistent data between the different methods is one of the major concerns and challenges for the pheno-genomic associations globally. Hopefully our dataset will provide additional value in this context.

line 216: REF? is reference missing?

Response: The reference is noted at the end of the sentence and the typo corrected.

line 224: Incomplete phrase missing information here: "Additional mutations in genes with- # were detected in <90% of reads".

Response: ” #” represents a superscript, and the following genes listed in the table have been observed in less than 90% of the sequencing reads, eg. Glu360*, Ile67fs, etc.

lines 254, 255, 268: ul or ug?

Response: corrected.

Table 2: please mention BDQ MIC (ug/ml) in the MIC column.

Response: Done.

Table 2: please provide reference for e MIC cut-off values for all different drugs mentioned here.

Response: We corrected to “MIC BDQ”, as the MICs were only measured for bedaquiline.

Line 320: do you mean the role of "Rv0678" and not "BDQ' mutations in poor treatment outcome?

Response: This sentence refers to the influence of BDQ resistance on poor treatment outcomes in general, following the example of Rv0678.

Fig. 1: please change to N=, as per universal standards. mentioning N36, N15 etc., gives the impression of sample number rather than total numbers.

Also please reconsider this figure design. it is understandable but the standard flow chart probably will be better.

Response: Thank you for the suggestions which we incorporated.

Fig. 2 resolution is poor. Can`t read anything on the X-axis.

Response: we increased the resolution.

Reviewer #2

The manuscript by Maghradze et al is a useful contribution to the field. It is well written, easy to follow and draws sound conclusions. Methods seem appropriate for the study. I only have some brief comments.

I wouldn’t say the main resistance mechanism in vitro is atpE. Whilst these mutants can be isolated, several studies report and over representation of rv0678 mutants as being the dominant mechanism in vitro. Maybe reporting that atpE is high level resistance is relevant?

Response: The main resistance mechanism in case of atpE, in the manuscript was considered as the biologically direct mechanism, which is specifically influencing the drug target. However, the majority of resistant cases globally are exhibiting mutations in Rv0678. We rephrased the sentenced for clarity.

In figure 2, maybe an additional indicating whether the strain is susceptible, intermediate or resistant to bedaquiline. There is the MIC data, and the are described in the text, but may make it easier to read.

Response: We tried to include in fig. 2 as much information as it was capable to provide. Drug resistance profiling gives opportunity to define resistance to first and second line medications, in addition to bdq MIC data. However, we tried to incorporate your recommendation and stratified resistance/susceptibility to bedaquiline using more contrasted colors for MIC column.

There are 9 strains in table 1 with MIC of 0.25 or 0.5, yet 12 mentioned in the text variants are identified. Of the three with no SNPs in target genes was WGS performed?

Response: The three isolates which exhibited MIC 0.25 or 0.5 had no mutations observed in the target genes, and yes, WGS was also performed on them– line 235.

The link to patient data is really nice, showing the influence of these mutations on outcomes. I think the statement “If confirmed, this would be similar to the situation with katG, for which the mutation Ser315Thr is rarely observed in the laboratory due to its high fitness cost in vitro despite being, by far, the most clinically relevant isoniazid resistance-conferring mutation due to its high fitness in vivo” is misleading, and should be altered or removed. I would argue that the disconnect with S315T is because in the lab any LOF mutation will provide resistance katG yet these mutants are non-viable in host due to the loss of catalase function. S315T is predominant in vivo, because it is one of the few mutations that preserves catalase function whilst also preventing the activation of INH. Again, also see comment above about frequency of atpE mutations.

Response: Noted and corrected.

Reviewer #3

Bedaquiline (BDQ) is a novel antituberculosis agent, that is pivotal to the potent multidrug regimens for the treatment of drug-resistant tuberculosis (TB). Although the genes associated with BDQ resistance has been described, the mutations that are associated with increased minimum inhibitory concentrations (MICs) have not been fully characterized. As such, the manuscript by Maghradze et al., that describes genomic correlates of BDQ resistance, provides useful information of interest to the field. Overall, the methods and results are presented clearly. A few detailed comments are bulleted below:

• Small inconsistencies in reporting number of programmatically determined BDQ-R isolates that were reclassified as susceptible or borderline:

o In the Results section, reports 3/15 initially classified as resistant that had a MIC<0.06 and no mutations, resulting in them being reclassified as susceptible. Also reports 2/15 initially classified as phenotypically resistant with MIC=0.12 and no resistance mutations. So should be 5/15 (?)

o In Discussion reports 6/15 identified through phenotypic testing turned out to be BDQ susceptible “based on repeated testing using both... pDST, as well as based on the genotypic data”. Unclear where this 6th isolate comes from. Presumably it is the one with a mutation at 4% allele frequency, but they specifically did not include AF in their inclusion criteria for mutations, so this is incredibly unclear.

Response: Six out of the fifteen isolates exhibited either borderline or no resistance. The numbers provided in the discussion section are correct; however, potential confusion might have been arisen from not explicitly stating that three isolates previously identified as phenotypically resistant exhibited an MIC of 0.12. Among these, one isolate harbored the Glu163Asp substitution in Rv0678 at a 4% allele frequency. For the clarification, we added an explanatory sentence in line 222.

• 3 isolates found to have MIC=0.25-0.5 (classified as phenotypically resistant) but no mutation in target genes

o They only look in the target genes atpE, pepQ, Rv0678. While these are the genes previously associated with resistance, I’m curious as to why they didn’t choose to investigate mutations in other genes associated with the MOA of bedaquiline or with the efflux pump.

Response: Our study aimed to investigate all detected variants in three key target genes (including mmpR), which have been identified as primary sources of resistance in previous studies and the WHO mutation catalog. The exploration of additional mechanisms of action contributing to bedaquiline resistance is the focus of our up-coming study.

• I understand why they would choose to only show treatment outcomes for those classified as phenotypically BDQR, however I think it could have strengthened the paper to demonstrate the difference in treatment outcomes between the resistance groups. In addition, as the authors note, the study and sample size preclude any meaningful discussion of clinical outcomes.

Response: Indeed, comparison of different MIC group outcomes would have been significant advantage. However, additional epidemiological data was not available for all patients, as long as our dataset included isolates harboring mutations in the target genes from 2008-20018, whilst routine pDST for bedaquiline was implemented only from 2019 onwards. We included the treatment outcome data for the patients treated from 2019 to the extent of availability.

• States that six of the phenotypically BDQR patients had a previous TB episode, but reports the percentage as 46.2% (of 12 total patients, 6 would be 50%). Then in the table, only lists 5 patients with known TB history and 1 with missing data.

Response: Corrected, five patients (41.7%) had experienced previous TB episode.

• Unclear if the 3 patients without known outcomes are still undergoing treatment, or if the data was truly unavailable, making it difficult to understand the cure rate.

Response: Table 2 shows available epidemiological data for the patients with MIC ≥0.25µg/mL, while the fields with “-“ is the information which was not accessible (line 279). In sense of the treatment outcomes, missing information means that the patient has left the country or was lost to follow-up.

• Included mixed-lineage isolates in their analyses and tree. This might not be as big of an issue for the BDQR mutation calling since they include multiple mutations at different allele frequencies for a few of their samples, but they should not be in the tree.

Response: The main objective of our study was to examine the relationship between mutation profiles and pDST, including MIC assay, while incorporating lineage information. As we are not aiming to infer transmission dynamics or draw conclusions based on phylogenetic relationships, we included isolates with mixed lineages. We took your recommendation into consideration and removed above mentioned samples in the revised version of the figure.

---

## [Decision Letter · Decision Letter 1]

Linking genetic and phenotypic bedaquiline resistance in Mycobacterium tuberculosis strains from Georgia

PONE-D-24-38581R1

Dear Dr. Gagneux,

We’re pleased to inform you that your manuscript has been judged scientifically suitable for publication and will be formally accepted for publication once it meets all outstanding technical requirements.

Kind regards,

Atul Vashist, PhD

Academic Editor

PLOS ONE

Additional Editor Comments (optional):

Reviewers' comments:

Reviewer's Responses to Questions

**Comments to the Author**

1. If the authors have adequately addressed your comments raised in a previous round of review and you feel that this manuscript is now acceptable for publication, you may indicate that here to bypass the “Comments to the Author” section, enter your conflict of interest statement in the “Confidential to Editor” section, and submit your "Accept" recommendation.

Reviewer #1: All comments have been addressed

Reviewer #2: All comments have been addressed

Reviewer #3: All comments have been addressed

2. Is the manuscript technically sound, and do the data support the conclusions?

Reviewer #1: Yes

Reviewer #2: Yes

Reviewer #3: Yes

3. Has the statistical analysis been performed appropriately and rigorously? 

Reviewer #1: N/A

Reviewer #2: Yes

Reviewer #3: N/A

4. Have the authors made all data underlying the findings in their manuscript fully available?

Reviewer #1: Yes

Reviewer #2: Yes

Reviewer #3: Yes

5. Is the manuscript presented in an intelligible fashion and written in standard English?

Reviewer #1: Yes

Reviewer #2: Yes

Reviewer #3: Yes

6. Review Comments to the Author

Reviewer #1: The authors have considered most of the suggestions and have answered the concerns raised by the reviewers to my satisfaction. they have presented a rebuttal that due to small sample size statistical analysis was not performed. I do not have the right expertise in the stastics to make any further comments.

I am happy to recommend this manuscript for publication in PlosOne.

Reviewer #2: The authors have addressed my previous comments. The manuscript will be a a valuable contribution to the field.

Reviewer #3: I do not have any further comments. The authors have addressed all my concerns from the original submission.

7. PLOS authors have the option to publish the peer review history of their article (what does this mean? ). If published, this will include your full peer review and any attached files.

**Do you want your identity to be public for this peer review?** For information about this choice, including consent withdrawal, please see our Privacy Policy .

Reviewer #1: No

Reviewer #2: No

Reviewer #3: No

---

## [Editor Report · Acceptance letter]

PONE-D-24-38581R1

PLOS ONE

Dear Dr. Gagneux,

I'm pleased to inform you that your manuscript has been deemed suitable for publication in PLOS ONE. Congratulations! Your manuscript is now being handed over to our production team.

Kind regards,

on behalf of

Dr. Atul Vashist

Academic Editor

PLOS ONE